# Hypermodels for Exploration

**Vikranth Dwaracherla, Xiuyuan Lu, Morteza Ibrahimi, Ian Osband, Zheng Wen, Benjamin Van Roy** [*]

## Abstract

We study the use of hypermodels to represent epistemic uncertainty and guide exploration. This generalizes and extends the use of ensembles to approximate Thompson sampling. The computational cost of training an ensemble grows with its size, and as such, prior work has typically been limited to ensembles with tens of elements. We show that alternative hypermodels can enjoy dramatic efficiency gains, enabling behavior that would otherwise require hundreds or thousands of elements, and even succeed in situations where ensemble methods fail to learn regardless of size. This allows more accurate approximation of Thompson sampling as well as use of more sophisticated exploration schemes. In particular, we consider an approximate form of information-directed sampling and demonstrate performance gains relative to Thompson sampling. As alternatives to ensembles, we consider linear and neural network hypermodels, also known as hypernetworks. We prove that, with neural network base models, a linear hypermodel can represent essentially any distribution over functions, and as such, hypernetworks are no more expressive.

## 1 Introduction

Consider the sequential decision problem of an agent interacting with an uncertain environment, aiming to maximize cumulative rewards. Over each time period, the agent must balance between exploiting existing knowledge to accrue immediate reward and investing in exploratory behavior that may increase subsequent rewards. In order to select informative exploratory actions, the agent must have some understanding of what it is uncertain about. As such, an ability to represent and resolve epistemic uncertainty is a core capability required of the intelligent agents.

The efficient representation of epistemic uncertainty when estimating complex models like neural networks presents an important research challenge. Techniques include variational inference (Blundell et al., 2015), dropout[1] (Gal & Ghahramani, 2016) and MCMC (Andrieu et al., 2003). Another approach has been motivated by the nonparametric bootstrap (Efron & Tibshirani, 1994) and trains an ensemble of neural networks with random perturbations applied to each dataset (Lu & Van Roy, 2017). The spirit is akin to particle filtering, where each element of the ensemble approximates a sample from the posterior and variation between models reflects epistemic uncertainty. Ensembles have proved to be relatively effective and to address some shortcomings of alternative posterior approximation schemes (Osband et al., 2016; 2018).

When training a single large neural network is computationally intensive, training a large ensemble of separate models can be prohibitively expensive. As such, ensembles in deep learning have typically been limited to tens of models (Riquelme et al., 2018). In this paper, we show that this parsimony can severely limit the quality of the posterior approximation and ultimately the quality of the learning system. Further, we consider more general approach based on *hypermodels* that can realize the benefits of large ensembles without the prohibitive computational requirements.

A *hypermodel* maps an index drawn from a reference distribution to a *base model*. An ensemble is one type of hypermodel; it maps a uniformly sampled base model index to that independently trained base model. We will consider additional hypermodel classes, including *linear hypermodels*, which we will use to map a Gaussian-distributed index to base model parameters, and *hypernetworks*, for which the mapping is a neural network (Ha et al., 2016). Our motivation is that intelligent

---

[*]DeepMind

[1]Although later work suggests that this dropout approximation can be of poor quality (Osband, 2016; Hron et al., 2017).

hypermodel design might be able to amortize computation across the entire distribution of base models, and in doing so, offer large gains in computational efficiency.

We train our hypermodels to estimate a posterior distribution over base models conditioned on observed data, in a spirit similar to that of the Bayesian hypermodel literature (Krueger et al., 2017). Unlike typical variational approximations to Bayesian deep learning, this approach allows computationally efficient training with complex multimodal distributions. In this paper, we consider hypermodels trained through stochastic gradient descent on perturbed data (see Section 2.1 for a full description). Training procedures for hypermodels are an important area of research, and it may be possible to improve on this approach, but that is not the focus of this paper. Instead, we aim to understand whether more sophisticated hypermodel architectures can substantially improve exploration. To do this we consider bandit problems of varying degrees of complexity, and investigate the computational requirements to achieve low regret over a long horizon.

To benchmark the quality of posterior approximations, we compare their efficacy when used for Thompson sampling (Thompson, 1933; Russo et al., 2018). In its ideal form, Thompson sampling (TS) selects each action by sampling a model from the posterior distribution and optimizing over actions. For some simple model classes, this approach is computationally tractable. Hypermodels enable *approximate* TS in complex systems where exact posterior inference is intractable.

Our results address three questions:

**Q: Can alternative hypermodels outperform ensembles?**
**A: Yes.** We demonstrate through a simple example that linear hypermodels can offer dramatic improvements over ensembles in the computational efficiency of approximate TS. Further, we demonstrate that linear hypermodels can be effective in contexts where ensembles fail regardless of ensemble size.

**Q: Can alternative hypermodels enable more intelligent exploration?**
**A: Yes.** We demonstrate that, with neural network hypermodels, a version of information-directed sampling (Russo & Van Roy, 2014; 2018) substantially outperforms TS. This exploration scheme would be computationally prohibitive with ensemble hypermodels but becomes viable with a hypernetwork.

**Q: Are hypernetworks warranted?**
**A: Not clear.** We prove a theorem showing that, with neural network base models, linear hypermodels can already represent essentially any distribution over functions. However, it remains to be seen whether hypernetworks can offer statistical or computational advantages.

Variational methods offer an alternative approach to approximating a posterior distribution and sampling from it. O'Donoghue et al. (2018) consider such an approach for approximating Thompson sampling in reinforcement learning. Approaches to approximating TS and information-directed sampling (IDS) with neural networks base models have been studied in (Lu & Van Roy, 2017; Riquelme et al., 2018) and Nikolov et al. (2019), respectively, using ensemble representations of uncertainty. Hypermodels have been a subject of growing interest over recent years. Ha et al. (2016) proposed the notion of hypernetworks as a relaxed form of weight-sharing. Krueger et al. (2017) proposed Bayesian hypernetworks for estimation of posterior distributions and a training algorithm based on variational Bayesian deep learning. A limitation of this approach is in its requirement that the hypernetwork be invertible. Karaletsos et al. (2018) studied Bayesian neural networks with correlated priors, specifically considering prior distributions in which units in the neural network are represented by latent variables and weights between units are drawn conditionally on the values of those latent variables. Pawlowski et al. (2017) introduced another variational inference based algorithm that interprets hypernetworks as implicit distributions, i.e. distributions that may have intractable probability density functions but allow for easy sampling. Hu et al. (2018) proposes the Stein neural sampler which samples from a given (un-normalized) probability distribution with neural networks trained by minimizing variants of Stein discrepancies.

## 2 HYPERMODEL ARCHITECTURES AND TRAINING

We consider base models that are parameterized by an element $\theta$ of a parameter space $\Theta$. Given $\theta \in \Theta$ and an input $X_t \in \Re^{N_x}$, a base model posits that the conditional expectation of the output

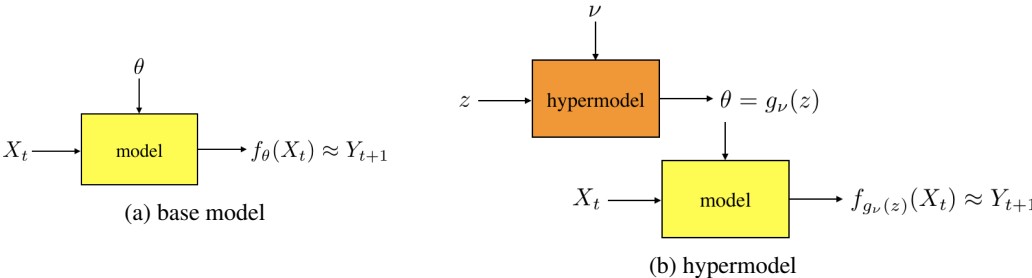

Figure 1: A base model generates an output $Y_t$ given parameters $\theta$ and input $X_t$, while a hypermodel generates base model parameters $g_\nu(z)$ given hypermodel parameters $\nu$ and an index $z$.

$Y_{t+1} \in \Re$ is given by $\mathbb{E}[Y_{t+1}|X_t, \theta] = f_\theta(X_t)$, for some class of functions $f$ indexed by $\theta$. Figure 1a depicts this class of parameterized base models.

A hypermodel is parameterized by parameters $\nu$, which identify a function $g_\nu : \mathcal{Z} \mapsto \Theta$. We will refer to each $z \in \mathcal{Z}$ as an index, as it identifies a specific instance of the base model. In particular, given hypermodel parameters $\nu$, base model parameters $\theta$ can be generated by selecting $z \in \mathcal{Z}$ and setting $\theta = g_\nu(z)$. This notion of a hypermodel is illustrated in Figure 1b. Along with a hypermodel, in order to represent a distribution over base models, we must specify a reference distribution $p_z$ that can be used to sample an element of $\mathcal{Z}$. A hypermodel and reference distribution together represent a distribution over base models through offering a mechanism for sampling them by sampling an index and passing it through the mapping.

## 2.1 HYPERMODEL TRAINING

Given a set of data pairs $\{(X_t, Y_{t+1}) : t = 0, \ldots, T-1\}$, a hypermodel training algorithm computes parameters $\nu$ so that the implied distribution over base model parameters approximates its posterior. It is important that training algorithms be incremental. This enables scalability and also allows for ongoing modifications to the data set, as those occurring in the bandit learning context, in which data samples accumulate as time progresses.

One approach to incrementally training a hypermodel involves perturbing data by adding noise to response variables, and then iteratively updating parameters via stochastic gradient descent. We will assume here that the reference distribution $p_z$ is either an $N_z$-dimensional unit Gaussian or a uniform distribution over the $N_z$-dimensional unit hypersphere. Consider an augmented data set $\mathcal{D} = \{(X_t, Y_{t+1}, A_t) : t = 0, \ldots, T-1\}$, where each $A_t \in \Re^{N_z}$ is a random vector that serves to randomize computations carried out by the algorithm. Each vector $A_t$ is independently sampled from $N(0, I)$ if $p_z$ is uniform over the unit hypersphere. Otherwise, $A_t$ is independently sampled from the unit hypersphere.

We consider a stochastic gradient descent algorithm that aims to minimize the loss function

$$\mathcal{L}(\nu, \mathcal{D}) = \int_{z \in \Re^{N_z}} p_z(dz) \left( \frac{1}{2\sigma_w^2} \sum_{(x,y,a) \in \mathcal{D}} (y + \sigma_w a^\top z - f_{g_\nu(z)}(x))^2 + \frac{1}{2\sigma_p^2} \|g_\nu(z) - g_{\nu_0}(z)\|_2^2 \right),$$

where $\nu_0$ is the initial vector of hypermodel parameters. Each iteration of the algorithm entails calculating the gradient of terms summed over a minibatch of $(x, y, a)$ tuples and random indices $z$. Note that $\sigma_w a^\top z$ here represents a random Gaussian perturbation of the response variable $y$. In particular, in each iteration, a minibatch $\tilde{\mathcal{D}}$ is constructed by sampling a subset of $\mathcal{D}$ uniformly with replacement, and a set $\tilde{\mathcal{Z}}$ of indices is sampled i.i.d. from $p_z$. An approximate loss function

$$\tilde{\mathcal{L}}(\nu, \tilde{\mathcal{D}}, \tilde{\mathcal{Z}}) = \frac{1}{|\tilde{\mathcal{Z}}|} \sum_{z \in \tilde{\mathcal{Z}}} \left( \frac{1}{2\sigma_w^2} \frac{|\mathcal{D}|}{|\tilde{\mathcal{D}}|} \sum_{(x,y,a) \in \tilde{\mathcal{D}}} (y + \sigma_w a^\top z - f_{g_\nu(z)}(x))^2 + \frac{1}{2\sigma_p^2} \|g_\nu(z) - g_{\nu_0}(z)\|_2^2 \right)$$

is defined based on these sets. Hypermodel parameters are updated according to $\nu \leftarrow \nu - \alpha \nabla_\nu \tilde{\mathcal{L}}(\nu, \tilde{\mathcal{D}}, \tilde{\mathcal{Z}})/|\mathcal{D}|$ where $\alpha$, $\sigma_w^2$, and $\sigma_p^2$ are algorithm hyperparameters. In our experiments, we

will take the step size $\alpha$ to be constant over iterations. It is natural to interpret $\sigma_p^2$ as a prior variance, as though the prior distribution over base model parameters is $N(0, \sigma_p^2 I)$, and $\sigma_w^2$ as the standard deviation of noise, as though the error distribution is $Y_t - f_\theta(X_t)|\theta \sim N(0, \sigma_w^2)$. Note, though, that a hypermodel can be trained on data generated by *any* process. One can think of the hypermodel and base models as inferential tools in the mind of an agent rather than a perfect reflection of reality.

## 2.2 Ensemble Hypermodels

An *ensemble hypermodel* is comprised of an ensemble of $N_\nu$ base models, each identified by a parameter vector in $\Theta = \Re^{N_\theta}$. Letting indices $\mathcal{Z}$ be the set of $N_\nu$-dimensional one-hot vectors, we can represent an ensemble in terms of a function $g_\nu : \mathcal{Z} \mapsto \Theta$ with parameters $\nu \in \Theta^{N_\nu}$. In particular, given hypermodel parameters $\nu \in \Theta^{N_\nu}$, an index $z \in \mathcal{Z}$ generates base model parameters $g_\nu(Z) = \nu Z$. For an ensemble hypermodel, the reference distribution $p_z$ is taken to be uniform over the $N_\nu$ elements of $\mathcal{Z}$.

## 2.3 Linear Hypermodels

Suppose that $\Theta = \Re^{N_\theta}$ and $\mathcal{Z} = \Re^{N_z}$. Consider a linear hypermodel, defined by $g_\nu(z) = a + Bz$, where hypermodel parameters are given by $\nu = (a \in \Re^{N_\theta}, B \in \Re^{N_\theta \times N_z})$ and $z \in \mathcal{Z}$ is an index with reference distribution $p_z$ taken to be the unit Gaussian $N(0, I)$ over $N_z$-dimensional vectors. Such a hypermodel can be used in conjunction with any base model that is parameterized by a vector of real numbers.

The aforementioned linear hypermodel entails a number of parameters that grows with the product of the number $N_\theta$ of base model parameters and the index dimension $N_z$, since $B$ is a $N_\theta \times N_z$ matrix. This can give rise to onerous computational requirements when dealing with neural network base models. For example, suppose that we wish to model neural network weights as a Gaussian random vector. This would require an index of dimension equal to the number of weights, and the number of hypermodel parameters would become quadratic in the number of neural network weights. For a large neural network, storing and updating that many parameters is impractical. As such, it is natural to consider linear hypermodels in which the parameters $a$ and $B$ are linearly constrained. Such linear constraints can, for example, represent independence or conditional independence structure among neural network weights.

## 2.4 Neural Network Hypermodels

More complex hypermodels are offered by neural networks. In particular, consider the case in which $g_\nu$ is a neural network with weights $\nu$, taking $N_z$ inputs and producing $N_\theta$ outputs. Such a representation is alternately refered to as a *hypernetwork*. Let the reference distribution $p_z$ be the unit Gaussian $N(0, I)$ over $N_z$-dimensional vectors. As a special case, a neural network hypermodel becomes linear if there are no hidden layers.

## 2.5 Additive Prior Models

In order for our stochastic gradient descent algorithm to operate effectively, it is often important to structure the base model so that it is a sum of a *prior model*, with parameters fixed at initialization, and a *differential model*, with parameters that evolve while training. The idea here is for the prior model to represent a sample from a prior distribution and for the differential to learn the difference between prior and posterior as training progresses. This additive decomposition was first introduced in (Osband et al., 2018), which demonstrated its importance in training ensemble hypermodels with neural network base models using stochastic gradient descent. Without this decomposition, to generate neural networks that represent samples from a sufficiently diffuse prior, we would have to initialize with large weights. Stochastic gradient descent tends to train too slowly and thus becomes impractical if initialized in such a way.

We will consider a decomposition that uses neural network base models (including linear base models as a special case) though the concept is more general. Consider a neural network model class $\{\tilde{f}_{\tilde{\theta}} : \tilde{\theta} \in \tilde{\Theta}\}$ with $\tilde{\Theta} = \Re^{N_{\tilde{\theta}}}$, where the parameter vector $\tilde{\theta}$ includes edge weights and node biases. Let the index set $\mathcal{Z}$ be $\Re^{N_z}$. Let $D$ be a diagonal matrix for which each element is the prior standard

deviation of corresponding component of $\tilde{\theta}$. Let $B \in \Re^{N_{\tilde{\theta}} \times N_z}$ be a random matrix produced at initialization. We will take $\tilde{\theta} = DBz$ to be parameters of the prior (base) model. Note that, given an index $z \in \mathcal{Z}$, this generates a prior model $\tilde{f}_{\tilde{\theta}} = \tilde{f}_{DBz}$. When we wish to completely specify a prior model distribution, we will need to define a distribution for generating the matrix $B$ as well as a reference distribution $p_z$.

Given a prior model of the kind we have described, we consider a base model of the form $f_\theta(x) = \tilde{f}_{\tilde{\theta}}(x) + \hat{f}_{\hat{\theta}}(x)$, where $\{\hat{f}_{\hat{\theta}} : \hat{\theta} \in \hat{\Theta}\}$ is another neural network model class satisfying $\hat{f}_0 = 0$, and $\theta$ is the concatenation of $\tilde{\theta}$ and $\hat{\theta}$. With $\tilde{\theta} = DBz$, the idea is to compute parameters $\hat{\theta}$ such that $f_\theta = \tilde{f}_{DBz} + \hat{f}_{\hat{\theta}}$ approximates a sample from a posterior distribution, conditioned on data. As such, $\hat{f}_{\hat{\theta}}$ represents a difference between prior and posterior. This decomposition is motivated by the observation that neural network training algorithms are most effective if initialized with small weights and biases. If we initialize $\hat{\theta}$ with small values, the initial values of $\hat{f}_{\hat{\theta}}$ will be small, and in this regime, $f_\theta \approx \tilde{f}_{DBz}$, which is appropriate since an untrained base model should represent a sample from a prior distribution. In general, $\hat{\theta}$ is the output of a neural network $\hat{g}_{\hat{\nu}}$ taking the same input $z$ as the prior hypermodel $\tilde{\theta} = DBz$. As is discussed above, in the course of training, we will only update $\hat{\nu}$ while keeping $D$ and $B$ fixed.

## 3 EXPLORATION SCHEMES

Our motivation for studying hypermodels stems from their potential role in improving exploration methods. As a context for studying exploration, we consider bandit problems. In particular, we consider the problem faced by an agent making sequential decisions, in each period selecting an action $X_t \in \mathcal{X}$ and observing a response $Y_{t+1} \in \Re$. Here, the action set $\mathcal{X}$ is a finite subset of $\Re^{N_x}$ and $Y_{t+1}$ is interpreted as a reward, which the agent wishes to maximize.

We view the environment as a channel that maps $X_t$ to $Y_{t+1}$, and conditioned on $X_t$ and the environment, $Y_{t+1}$ is conditionally independent of $X_0, Y_1, \ldots, X_{t-1}, Y_t$. In other words, actions do not induce delayed consequences. However, the agent learns about the environment from applying actions and observing outcomes, and as such, its prediction of an outcome $Y_{t+1}$ is influenced by past observations $X_0, Y_1, \ldots, X_{t-1}, Y_t$.

A base model serves as a possible realization of the environment, while a hypermodel encodes a belief distribution over possible realizations. We consider an agent that represents beliefs about the environment through a hypermodel, continually updating hypermodel paramerers $\nu$ via stochastic gradient descent, as described in Section 2.1, to minimize a loss function based on past actions and observations. At each time $t$, the agent selects action $X_t$ based on the current hypermodel. Its selection should balance between exploring to reduce uncertainty indicated by the hypermodel and exploiting knowledge conveyed by the hypermodel to accrue rewards.

### 3.1 THOMPSON SAMPLING

TS is a simple and often very effective exploration scheme that will serve as a baseline in our experiments. With this scheme, each action $X_t$ is selected by sampling an index $z \sim p_z$ from the reference distribution and then optimizing the associated base model to obtain $X_t \in \arg\max_{x \in \mathcal{X}} f_{g_\nu(z)}(x)$. See (Russo et al., 2018) for an understanding when and why TS is effective.

### 3.2 INFORMATION-DIRECTED SAMPLING

IDS (Russo & Van Roy, 2014; 2018) offers an alternative approach to exploration that aims to more directly quantify and optimize the value of information. There are multiple versions of IDS, and we consider here a sample-based version of *variance-IDS* (Russo & Van Roy, 2018). In each time period, this entails sampling a new multiset $\tilde{\mathcal{Z}}$ i.i.d. from $p_z$. Then, for each action $x \in \mathcal{X}$ we compute the sample mean of immediate regret

$$r_x = \frac{1}{|\tilde{\mathcal{Z}}|} \sum_{z \in \tilde{\mathcal{Z}}} \left( \max_{x^* \in \mathcal{X}} f_{g_\nu(z)}(x^*) - f_{g_\nu(z)}(x) \right)$$

and a sample variance of reward across possible realizations of the optimal action

$$v_x = \sum_{x^* \in \mathcal{X}} \frac{|\tilde{\mathcal{Z}}_{x^*}|}{|\tilde{\mathcal{Z}}|} \left( \frac{1}{|\tilde{\mathcal{Z}}_{x^*}|} \sum_{z \in \tilde{\mathcal{Z}}_{x^*}} f_{g_\nu(z)}(x) - \frac{1}{|\tilde{\mathcal{Z}}|} \sum_{z \in \tilde{\mathcal{Z}}} f_{g_\nu(z)}(x) \right)^2 .$$

Here, $\{\tilde{\mathcal{Z}}_{x^*} : x^* \in \mathcal{X}\}$ forms a partition of $\tilde{\mathcal{Z}}$ such that, $x^*$ is an optimal action for each $z \in \tilde{\mathcal{Z}}_{x^*}$; that is, $\tilde{\mathcal{Z}}_{x^*} = \{z \in \tilde{\mathcal{Z}} | x^* \in \arg\max_{x \in \mathcal{X}} f_{g_\nu(z)}(x)\}$ Then, a probability vector $\pi^* \in \Delta_{\mathcal{X}}$ is obtained by solving

$$\pi^* \in \arg\min_{\pi \in \Delta_{\mathcal{X}}} \frac{\left( \sum_{x \in \mathcal{X}} \pi_x r_x \right)^2}{\sum_{x \in \mathcal{X}} \pi_x v_x},$$

and action $X_t$ sampled from $\pi^*$. Note that $\pi_x^* = 0$ if $\tilde{\mathcal{Z}}_x$ is empty. As established by (Russo & Van Roy, 2018), the minimum over $\pi \in \Delta_{\mathcal{X}}$ is always attained by a probability vector that has at most two nonzero components, and this fact can be used to simplify optimization algorithms.

Producing reasonable estimates of regret and variance calls for many distinct samples, and the number required scales with the number of actions. An ensemble hypermodel with tens of elements does not suffice, while alternative hypermodels we consider can generate very large numbers of distinct samples.

## 4 CAN HYPERMODELS OUTPERFORM ENSEMBLES?

Because training a large ensemble can be prohibitively expensive, neural network ensembles have typically been limited to tens of models (Riquelme et al., 2018). In this section, we demonstrate that a linear hypermodel can realize the benefits of a much larger ensemble without the onerous computational requirements.

### 4.1 GAUSSIAN BANDIT WITH INDEPENDENT ARMS

We consider a Gaussian bandit with $K$ independent arms where the mean reward vector $\theta^* \in \Re^K$ is drawn from a Gaussian prior $N(0, \sigma_p^2 I)$. During each time period $t$, the agent selects an action $X_t$ and observes a noisy reward $Y_{t+1} = \theta_{X_t}^* + W_{t+1}$, where $W_{t+1}$ is i.i.d. $N(0, \sigma_w^2)$. We let $\sigma_p^2 = 2.25$ and $\sigma_w^2 = 1$, and we fix the time horizon to 10,000 periods.

We compare an ensemble hypermodel and a diagonal linear hypermodel trained via SGD with perturbed data. Our simulation results show that a diagonal linear hypermodel requires about 50 to 100 times less computation than an ensemble hypermodel to achieve our target level of performance.

As discussed in Section 2.5, we consider base models of the form $f_\theta(x) = \tilde{f}_{DBz}(x) + \hat{f}_{\hat{\theta}}(x)$, where $\tilde{f}_{DBz}(x)$ is an additive prior model, and $\hat{f}_{\hat{\theta}}(x)$ is a trainable differential model that aims to learn the difference between prior and posterior. For an independent Gaussian bandit, $\tilde{f}_{\bar{\theta}}(x) = \hat{f}_{\bar{\theta}}(x) = \bar{\theta}_x$ for all $\bar{\theta}$ and x. Although the use of prior models is inessential in this toy example, we include it for consistency and illustration of the approach.

The index $z \in \Re^{N_z}$ of an ensemble hypermodel is sampled uniformly from the set of $N_z$-dimensional one-hot vectors. Each row of $B \in \Re^{K \times N_z}$ is sampled from $N(0, I)$, and $D = \sigma_p I$. The ensemble (differential) hypermodel takes the form $\hat{g}_{\hat{\nu}}(z) = \hat{\nu} z$, where the parameters $\hat{\nu} \in \Re^{K \times N_z}$ are initialized to i.i.d. $N(0, 0.05^2)$. Although initializing to small random numbers instead of zeros is unnecessary for a Gaussian bandit, our intention here is to mimic neural network initialization and treat the ensemble hypermodel as a special case of neural network hypermodels.

In a linear hypermodel, to model arms independently, we let $z_1, \ldots, z_K \in \Re^m$ each be drawn independently from $N(0, I)$, and let the index $z \in \Re^{N_z}$ be the concatenation of $z_1, \ldots, z_K$, with $N_z = Km$. Let the prior parameters $b_1, \ldots, b_K \in \Re^m$ be sampled uniformly from the $m$-dimensional hypershpere, and let $B \in \Re^{K \times N_z}$ be a block matrix with $b_1^\top, \ldots, b_K^\top$ on the diagonal and zero everywhere else. Let $D = \sigma_p I$. The diagonal linear (differential) hypermodel takes the form $\hat{g}_{\hat{\nu}}(z) = Cz + \mu$, where $\mu \in \Re^K$ and matrix $C \in \Re^{K \times N_z}$ has a block diagonal structure $C = \text{diag}(c_1^\top, \ldots, c_K^\top)$, with $c_1, \ldots, c_K \in \Re^m$. The hypermodel parameters $\hat{\nu} = (C, \mu)$ are initialized to i.i.d. $N(0, 0.05^2)$.

We train both hypermodels using SGD with perturbed data. For an ensemble hypermodel, the perturbation of the data point collected at time $t$ is $\sigma_w A_t^\top z$, where $A_t \sim N(0, I)$. For a diagonal linear hypermodel, the perturbation is $\sigma_w A_t^\top z_{X_t}$, where $A_t$ is sampled uniformly from the $m$-dimensional unit hypersphere.

We consider an agent to perform well if its average regret over 10,000 periods is below $0.01\sqrt{K}$. We compare the computational requirements of ensemble and diagonal linear hypermodels across different numbers of actions. As a simple machine-independent definition, we approximate the number of arithmetic operations over each time period:

$$\text{computation} = n_{\text{sgd}} \times n_z \times n_{\text{data}} \times n_{\text{params}},$$

where $n_{\text{sgd}}$ is the number of SGD steps per time period, $n_z$ is the number of index samples per SGD step, $n_{\text{data}}$ is the data batch size, and $n_{\text{params}}$ is the number of hypermodel parameters involved in each index sample. We fix the data batch size to 1024 for both agents, and sweep over other hyperparameters separately for each agent. All results are averaged over 100 runs. In Figure 2, we plot the computation needed versus number of actions. Using a diagonal linear hypermodel dramatically reduces the amount of computation needed to perform well relative to using an ensemble hypermodel, with a speed-up of around 50 to 100 times for large numbers of actions.

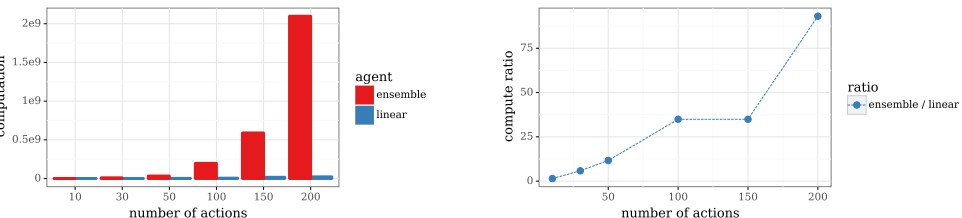

Figure 2: Computation required for ensemble hypermodels versus diagonal linear hypermodels to perform well on Gaussian bandits with independent arms.

### 4.2 NEURAL NETWORK BANDIT

In this section we show that linear hypermodels can also be more effective than ensembles in settings that require *generalization* between actions. We consider a bandit problem with rewards generated by a neural network that takes vector-valued actions as inputs. We consider a finite action set $\mathcal{A} \subset \Re^d$ with $d = 20$, sampled uniformly from the unit hypersphere. We generate data using a neural network with input dimension 20, 2 hidden layers of size 3, and a scalar output. The output is perturbed by i.i.d. $N(0, 1)$ observation noise. The weights of each layer are sampled independently from $N(0, 2.25)$, $N(0, 2.25/3)$, and $N(0, 2.25/3)$, respectively, with biases from $N(0, 1)$.

We compare ensemble hypermodels with 10, 30, 100, and 300 particles, and a linear hypermodel with index dimension 30. Both agents use an additive prior $\tilde{f}_{DBz}(x)$, where $\tilde{f}$ is a neural network with the same architecture as the one used to generate rewards. For the ensemble hypermodel, each row of $B$ is initialized by sampling independently from $N(0, I)$, and $D$ is diagonal with appropriate prior standard deviations. For the linear hypermodel, we enforce independence of weights across layers by choosing $B$ to be block diagonal with 3 blocks, one for each layer. Each block has width 10. Within each block, each row is initialized by sampling uniformly from the 10-dimensional unit hypersphere. For the trainable differential model, both agents use a neural network architecture with 2 hidden layers of width 10. The parameters of the ensemble hypermodel are initialized to truncated $N(0, 0.05^2)$. The weights of the linear hypermodel are initialized using the Glorot uniform initialization, while the biases are initialized to zero.

In our simulations, we found that training without data perturbation gives lower regret for both agents. In Figure 3, we plot the cumulative regret of agents trained without data perturbation. We see that linear hypermodels achieve the least regret in the long run. The performance of ensemble hypermodels is comparable when the number of actions is 200. However, there is a large performance gap when the number of actions is greater than 200, which, surprisingly, cannot be compensated by increasing the ensemble size. We suspect that this may have to do with the reliability of neural network regression, and linear hypermodels are somehow able to circumvent this issue.

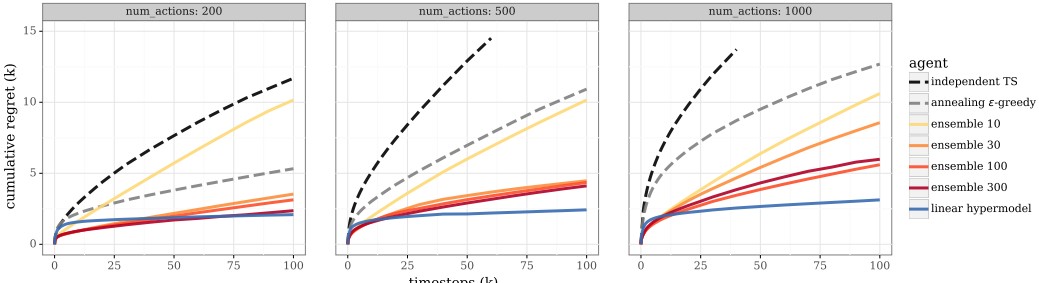

Figure 3: Compare (i) ensemble hypermodels, (ii) linear hypermodels, (iii) annealing $\epsilon$-greedy, and (iv) an agent assuming independent actions on a neural network bandit.

We also compare with an $\epsilon$-greedy agent with a tuned annealing rate, and an agent that assumes independent actions and applies TS under the Gaussian prior and Gaussian noise assumption. The gap between the $\epsilon$-greedy agent and hypermodel agents grows as the number of actions becomes large, as $\epsilon$-greedy explores uniformly and does not write off bad actions. The performance of the agent that assumes independent actions degrades quickly as the number of actions increases, since it does not generalize across actions. In the appendix, we also discuss Bayes by Backprop (Blundell et al., 2015) and dropout (Gal & Ghahramani, 2016) as approximation methods for posterior sampling.

## 5    CAN HYPERMODELS ENABLE MORE INTELLIGENT EXPLORATION?

IDS, as we described earlier, requires a large number of independent samples from the (approximate) posterior distribution to generate an action. One way to obtain these samples is to maintain an ensemble of models, as is done by Nikolov et al. (2019). However, as the number of actions increases, maintaining performance requires a large ensemble, which becomes computationally prohibitive. More general hypermodels offer an efficient mechanism for generating the required large number of base model samples. In this section, we present experimental results involving a problem and hypermodel stylized to demonstrate advantages of IDS in a transparent manner. This context is inspired by the one-sparse linear bandit problem constructed by Russo & Van Roy (2018). However, the authors of that work do not offer a general computationally practical approach that implements IDS. Hypermodels may serve this need.

We generate data according to $Y_{t+1} = X_t^\top \theta^* + W_{t+1}$ where $\theta^* \in \Re^{N_\theta}$ is sampled uniformly from one-hot vectors and $W_{t+1}$ is i.i.d. $N(0,1)$ noise. We consider a linear base model $f_\theta(x) = \theta^\top x$ and hypermodel $(g_\nu(z))_m = \exp(\beta \nu_m (z_m^2 + \alpha)) / \sum_{n=1}^{N_\theta} \exp(\beta \nu_n (z_n^2 + \alpha))$, where $\alpha = 0.01$, and $\beta = 10$. As a reference distribution we let $p_z$ be $N(0, I)$. Let the initial hypermodel parameters $\nu_0$ be the vector with each component equal to one. Note that our hypermodel is designed to allow representation of the prior distribution, as well as uniform distributions over subsets of one-hot vectors. For simplicity, let $N_\theta$ be a power of two. Let $\mathcal{I}$ be the set of indicator vectors for all non-singleton sublists of indices in $(1, \ldots, N_\theta)$ that can be obtained by bisecting the list one or more times. Note that $|\mathcal{I}| = N_\theta - 2$. Let the action space $\mathcal{X}$ be comprised one hot-vectors and vectors $\{x/2 : x \in \mathcal{I}\}$.

As with the one-sparse linear bandit of (Russo & Van Roy, 2018), this problem is designed so that TS will identify the nonzero component of $\theta^*$ by applying one-hot actions to rule out one component per period, whereas IDS will essentially carry out a bisection search. This difference in behavior stems from the fact that TS will only ever apply actions that have some chance of being optimal, which in this context includes only the one-hot vectors, whereas IDS can apply actions known to be suboptimal if they are sufficiently informative.

Figure 4 plots regret realized by TS and variance-IDS using the aforementioned hypermodel, trained with perturbed SGD. As expected, the difference in performance is dramatic. Each plot is averaged over 500 simulations. We used SGD hyperparameters $\sigma_w^2 = 0.01$ and $\sigma_p^2 = 1/\log N_\theta$. The experiments are with $N_\theta = 200$, 500 samples are used for computing the variance-based information ratio.

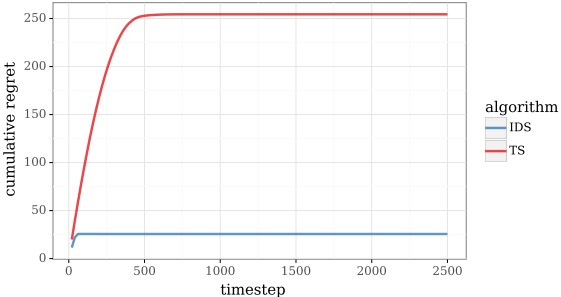

Figure 4: Cumulative regret of IDS and TS with with one-sparse models.

## 6 ARE HYPERNETWORKS WARRANTED?

Results of the previous sections were generated using ensemble and linear hypermodels. It remains to be seen whether hypernetworks offer substantial benefits. One might believe that hypernetworks can benefit from the computational advantages enjoyed by linear hypermodels while offering the ability to represent a broader range of probability distributions over base models. The following result refutes this possibility by establishing that, with neural network base models, linear hypermodels can represent essentially *any* probability distribution over functions with finite domain. We denote by $L_\infty(\mathcal{X}, B)$ the set of functions $f : \mathcal{X} \mapsto \Re$ such that $\|f\|_\infty < B$, where $\mathcal{X}$ is finite with $|\mathcal{X}| = K$.

**Theorem 1** *Let $p_z$ be the unit Gaussian distribution in $\Re^K$. For all $\epsilon > 0$, $\delta > 0$, $B > 0$, and probability measures $\mu$ over $L_\infty(\mathcal{X}, B)$, there exist a transport map $H$ from $p_z$ to $\mu$, a neural network $f_\theta : \mathcal{X} \mapsto \Re$ with a linear output node and ReLU hidden nodes, and a linear hypermodel $g_\nu : \mathcal{Z} \mapsto \Re^{N_\theta}$ with form $g_\nu(z) = \left[z^T, \nu^T\right]^T$ such that*

$$\|f_{g_\nu(z)} - f^*\|_\infty \leq \epsilon$$

*with probability at least $1 - \delta$, where $f^* = H(z)$.*

This result is established in Appendix A. To digest the result, first suppose that the inequality is satisfied with $\epsilon = 0$. Interpret $\mu$ as the target probability measure we wish to approximate using the hypermodel. Note that $f_{g_\nu(z)}$ and $f^*$ are determined by $z \sim p_z$, and $f^*$ is distributed according to $\mu$, since $H$ is a transport function that maps $p_z$ to $\mu$. If $\|f_{g_\nu(z)} - f^*\|_\infty = 0$ then $f_{g_\nu(z)}$ is also distributed according to $\mu$, and as such, the hypermodel perfectly represents the distribution. If $\epsilon > 0$, the representation becomes approximate with tolerance $\epsilon$.

Though our result indicates that linear hypermodels suffice to represent essentially all distributions over functions, we do not rule out the possibility of statistical or computational advantages to using hypernetworks. In particular, there could be situations where hypernetworks generalize more accurately given limited data, or where training algorithms operate more effectively with hypernetworks. In supervised learning, deep neural networks offer such advantages even though a single hidden layer suffices to represent essentially any function. Analogous benefits might carry over to hypernetworks, though we leave this question open for future work.

## 7 CONCLUSION

Our results offer initial signs of promise for the role of hypermodels beyond ensembles in improving exploration methods. We have shown that linear hypermodels can offer large gains in computational efficiency, enabling results that would otherwise require ensembles of hundreds or thousands of elements. Further, these efficiency gains enable more sophisticated exploration schemes. In particular, we experiment with a version of IDS sampling and demonstrate benefits over methods based on TS. Finally, we consider the benefits of hypernetworks and establish that, with neural network base models, linear hypermodels are already able to represent essentially any distribution over functions. Hence, to the extent that hypernetworks offer advantages, this would not be in terms of the class of distributions that can be represented.

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

## A  Universal Approximation via Linear Hypermodels

Assume that $\mathcal{X} \subset \Re$ is a finite set with $|\mathcal{X}| = K$, and $\mu$ is a probability measure over the bounded functions $f : \mathcal{X} \to \Re$ such that $\|f\|_\infty \leq B$. First, we show that we can approximately sample a function from $\mu$ using a ReLU model with an linear hypermodel, with the input to the hypermodel drawn from the $K$-dimensional uniform distribution. Our main result is summarized below:

**Theorem 2** *Let $p_z$ be the uniform distribution over $[0,1]^K$. For all $\epsilon > 0$, $\delta \in (0,1)$, $B > 0$, and probability measures $\mu$ over $L_\infty(\mathcal{X}, B)$, there exists a transport map $H$ from $p_z$ to $\mu$, a neural network $f_\theta : \mathcal{X} \mapsto \Re$ with a linear output node and ReLU hidden nodes, and a linear hypermodel $g_\nu : \mathcal{Z} \mapsto \Re^{N_\theta}$ with form $g_\nu(z) = \left[z^T, \nu^T\right]^T$ such that*

$$\|f_{g_\nu(z)} - f^*\|_\infty \leq \epsilon,$$

*with probability at least $1 - \delta$, where $f^* = H(z)$.*

**Proof of Theorem 2:** Note that since $|\mathcal{X}| = K$, the functions $f : \mathcal{X} \to \Re$ can be represented as vectors in $\Re^K$ and hence $\mu$ can be viewed as a probability measure over $\Re^K$. Since $p_z$ is absolutely continuous with respect to the Lebesgue measure, from Brenier's theorem, there exists a measurable transport map $H : \Re^K \to \Re^K$ from $p_z$ to $\mu$. Notice that we can always assume $H(z) = 0$ for $z \notin [0,1]^K$ since all the probability mass under $p_z$ is in $[0,1]^K$, and this assumption does not affect the measurability of $H$. To show that each component of $H$ is Lebesgue integrable, let $H(z)[x]$ denote the component of $H(z)$ corresponding to $x$. Note that

$$\int_{\Re^K} |H(z)[x]| dz = \int_{[0,1]^K} |H(z)[x]| dz \leq \int_{[0,1]^K} B dz = B,$$

where the inequality follows from the fact that $\mu$ is over $L_\infty(\mathcal{X}, B)$.

From Theorem 1 in Lu et al. (2017), for any $\epsilon > 0$ and $\delta \in (0,1)$, there exists a ReLU model $\tilde{H} : \Re^K \to \Re^K$ s.t. for any $x \in \mathcal{X}$,

$$\int_{\Re^K} |H(z)[x] - \tilde{H}(z)[x]| dz < \epsilon\delta/K,$$

where $H(z)[x]$ and $\tilde{H}(z)[x]$ are respectively the component of $H(z)$ and $\tilde{H}(z)$ corresponding to $x$. Note that the above inequality implies:

$$\int_{\Re^K} \|H(z) - \tilde{H}(z)\|_\infty dz \leq \int_{\Re^K} \|H(z) - \tilde{H}(z)\|_1 dz = \sum_{x \in \mathcal{X}} \int_{\Re^K} |H(z)[x] - \tilde{H}(z)[x]| dz < \epsilon\delta.$$

Hence we have

$$E_{z \sim p_z} \left[\|H(z) - \tilde{H}(z)\|_\infty\right] = \int_{[0,1]^K} \|H(z) - \tilde{H}(z)\|_\infty dz \leq \int_{\Re^K} \|H(z) - \tilde{H}(z)\|_\infty dz < \epsilon\delta.$$

Note that we can always assume $\|\tilde{H}(z)\|_\infty \leq B$ for $z \in [0,1]^K$. If this assumption does not hold, we can add some ReLU layers to cap $\tilde{H}$ to ensure $\|\tilde{H}(z)\|_\infty \leq B$. Since $\|H(z)\|_\infty \leq B$ almost surely, this cap will not increase $E_{z \sim p_z} \left[\|H(z) - \tilde{H}(z)\|_\infty\right]$.

We now discuss how to implement a ReLU model $\tilde{h} : \mathcal{X} \times [0,1]^K \to [-B, B]$ s.t. $\tilde{h}(x, z) = \tilde{H}(z)[x]$ based on the ReLU implementation of the $K$-dimensional $\tilde{H}(z)$. Note that it is straightforward to use ReLU to implement the $K$-dimensional one-hot encoding for all $x \in \mathcal{X}$. Since $\|\tilde{H}(z)\|_\infty \leq B$, by defining

$$\tilde{h}(x, z) = \left[\sum_{x' \in \mathcal{X}} \max\left\{4B \times \mathbf{1}(x' = x) + \tilde{H}(z)[x] - 2B, 0\right\}\right] - 2B$$

we have $\tilde{h}(x, z) = \tilde{H}(z)[x]$. Since both $\tilde{H}(z)$ and the one-hot encoding can be implemented by ReLU, $\tilde{h}(x, z)$ can also be implemented by ReLU.

Finally, we discuss how to construct the target ReLU model $f_\theta : \mathcal{X} \to \Re$ and the linear hypermodel $g_\nu$ based on $\tilde{h}$. Note that the ReLU corresponding to $\tilde{h}$ has $K+1$ input nodes, one corresponding to $x$ (called InputX) and $K$ corresponding to $z$ (called InputZ). Thus, by treating $z$ as part of the parameter vector $\theta$, $f_\theta : \mathcal{X} \to \Re$ is construct as follows: we make InputZ hidden nodes, and the only input node to each node in InputZ is InputX. Specifically, the input to the $i$th node in InputZ is $0 \times x + z_i$ with scalar weight 0 and bias $z_i$, where $z_i$ is the $i$th component of $z$. Also note that given $\tilde{h}$, the components in $\theta$ are either constant or components in $z$, thus $g_\nu$ is linear and can be written as $g_\nu(z) = \left[z^T, \nu^T\right]^T$. Since the components in $z$ are statistically independent, and components in $\theta = g_\nu(z)$ are either constant or components in $z$, consequently, the components in $\theta$ are statistically independent. Note that by definition, $f_{g_\nu(z)}(x) = \tilde{h}(x, z) = \tilde{H}(z)[x]$. By defining $f^* = H(z)$, we have $E_{z \sim p_z}\left[\|f_{g_\nu(z)} - f^*\|_\infty\right] < \epsilon\delta$. From Markov's, with probability at least $1 - \delta$, we have $\|f_{g_\nu(z)} - f^*\|_\infty \le \epsilon$. **q.e.d.**

Finally, we prove Theorem 1 based on Theorem 2:

**Proof of Theorem 1:** The proof is similar to that of Theorem 2. Recall that $\mu$ can be viewed as a probability measure over $\Re^K$. Since $p_z = N(0, I_K)$ is absolutely continuous with respect to the Lebesgue measure, from Brenier's theorem, there exists a measurable transport map $H : \Re^K \to \Re^K$ from $p_z$ to $\mu$, moreover, $H(z) = \nabla_z \phi(z)$ for a convex scalar function $\phi : \Re^K \to \Re$. Notice that for the given $\delta \in (0, 1)$, we can always choose a large enough $\eta > 0$, such that $P(z \in \mathcal{B}_\eta) \ge 1 - \delta/2$, where $\mathcal{B}_\eta = \{z : \|z\|_2 \le \eta\}$.

We define an auxiliary function $H' : \Re^K \to \Re^K$ as

$$H'(z) = \begin{cases} H(z) & \text{if } z \in \mathcal{B}_\eta \\ 0 & \text{otherwise} \end{cases}$$

Since $H$ is measurable, $H'$ is also measurable. Moreover, since $H'(z) = H(z) = \nabla_z \phi(z)$ on the compact set $\mathcal{B}_\eta$, thus $H'(z)$ is bounded in $\mathcal{B}_\eta$. Thus, for any $x \in \mathcal{X}$, we have

$$\int_{\Re^K} |H'(z)[x]|\, dz = \int_{\mathcal{B}_\eta} |H'(z)[x]|\, dz < \infty.$$

Similar to the proof for Theorem 2, for the given $\epsilon$ and $\delta$, there exists a ReLU model $\tilde{H} : \Re^K \to \Re^K$ s.t.

$$\int_{\Re^K} \|H'(z) - \tilde{H}(z)\|_\infty dz < \epsilon\delta/2.$$

Notice that

$$E_{z \sim p_z}\left[\|H'(z) - \tilde{H}(z)\|_\infty\right] = \int_{\Re^K} \|H'(z) - \tilde{H}(z)\|_\infty p_z(z) dz < \int_{\Re^K} \|H'(z) - \tilde{H}(z)\|_\infty dz < \epsilon\delta/2,$$

where with a little bit abuse of notation $p_z(\cdot)$ denotes the probability density function of the probability measure $p_z$. The first inequality follows from $p_z(z) \le (2\pi)^{-K/2} < 1$.

The subsequent analysis is similar to that in Theorem 2. Similarly, we can prove that with probability at least $1 - \delta/2$, we have $\|f_{g_\nu(z)} - H'(z)\|_\infty \le \epsilon$, where $f_\theta$ is a ReLU model and $g_\nu$ is a linear hypermodel and can be written as $g_\nu(z) = \left[z^T, \nu^T\right]^T$. Moreover, the components in $\theta = g_\nu(z)$ are statistically independent. Recall that with probability at least $1 - \delta/2$, we have $H'(z) = H(z)$. Thus, from the union bound, we have with probability at least $1 - \delta$, we have $\|f_{g_\nu(z)} - H(z)\|_\infty \le \epsilon$. By setting $f^* = H(z)$, we have proved the theorem. **q.e.d.**

## B  ADDITIVE PRIORS

Osband et al. (2018) discuss the benefits of using an additive prior to represent prior uncertainty over models. In sequential decision making, it is particularly crucial to be able to represent prior uncertainties when there is little data available. In this section, we demonstrate the effectiveness of additive priors by comparing ensembles with and without additive priors on the neural network bandit problem described in Section 4.2. Both ensembles are initialized randomly using standard neural network initializations. Since weights are typically initialized to small values (otherwise

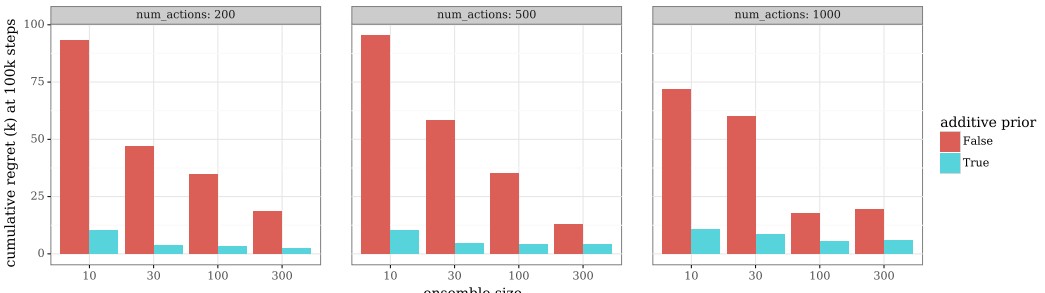

Figure 5: Compare ensembles with and without additive priors on a neural network bandit.

training could be difficult), the outputs of ensembles that do not use additive priors will be close to zero and will not reflect prior uncertainties in general. We see in Figure 5 that ensembles with additive priors achieve significantly lower regrets across different numbers of actions and ensemble sizes.

## C    ALGORITHM SENSITIVITY ANALYSIS

In order to analyze the sensitivity of the algorithm on different parameters, we present a series of experiments on the neural network bandit, similar to Section 4.2. The default values of the parameters used in these experiments are same as the values used to generate Figure 3. The results from the experiments are presented below.

In Figure 6 we present some results from sensitivity analysis on observation noise and perturbation noise. The plot shows the regret of linear hypermodel at 100k steps, with different perturbation scales $\tilde{\sigma}_w$ (standard deviation of the noise added to the response variable in the loss function) and standard deviations of the observation noise $\sigma_w$. Both $\tilde{\sigma}_w$ and $\sigma_w$ take values from $\{0, 1, 2\}$. Observe that for $\sigma_w = 0$ and $\sigma_w = 1$, perturbation scale of 0 works the best; however, on increasing the observation noise to $\sigma_w = 2$, a perturbation scale of 1 performs better than perturbation scale of 0. The reason for this could be that SGD step is introducing sufficient amount of noise when $\sigma_w$ is 0 or 1, but it seems that we need to inject additional noise for larger observation noise. From this, it is clear that we need to introduce perturbation into the loss function, as the observation noise grows. Even though there is a discrepancy in the cumulative regret for different perturbation scales, the algorithm seems to be robust to the variation in perturbation scale across different levels of observation noise.

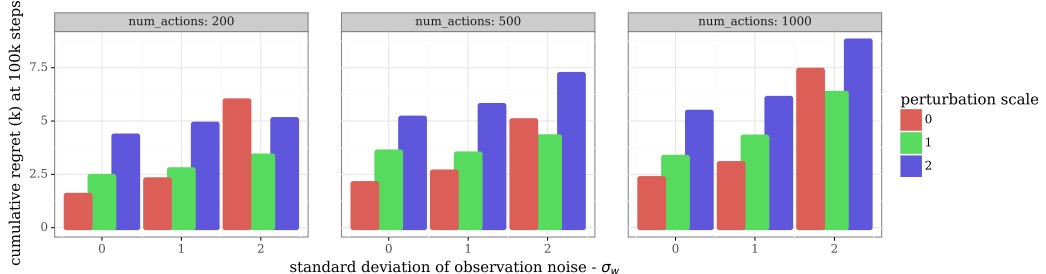

Figure 6: Performance of linear hypermodel with varying strengths in observation noise and perturbation

In Figure 7 we present results on how mis-specification in prior can affect the performance of the linear hypermodel. Plot shows the regret of linear hypermodel after 100k steps, when the prior is mis-specified. Prior is mis-speicified by drawing weights of the prior network from a distribution such that the variance of these weights are a factor $m$ times that of the variance of the weights of the generator, we call this value $m$ as the prior weight multiplier. We can see that a very small value

of $m$ does not induce sufficient exploration and leads to a huge regret. Similarly, a large value of $m$ can also induce more exploration than desired and leads to some additional regret.

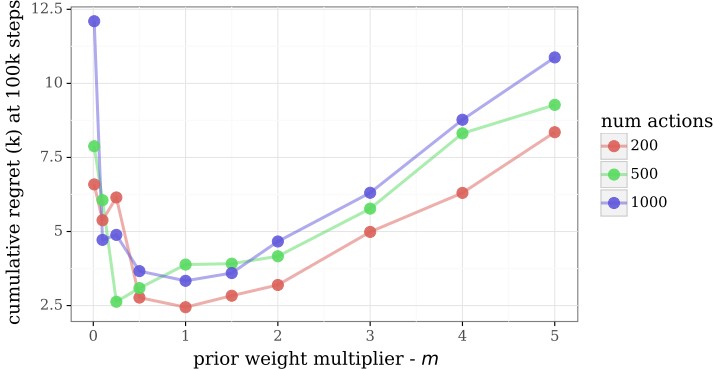

Figure 7: Performance of linear hypermodels for different values of multiplier $m$

In Figure 8, we show how performance of a linear hypermodel is affected by the index dimension. Recall from Section 4.2 that we use a disjoint segment of the index vector to generate prior weights for each layer. We vary the index dimension per layer as 1, 2, 3, 5 and 10 (corresponding to the entire index vector with dimension 3, 6, 9, 15, and 30) for 500 random seeds, and observe the average cumulative regret attained by the algorithm after 100k steps. Although there is some noise, we observe that as the index dimension increases the cumulative regret decreases. However, the improvement is marginal beyond an index dimension.

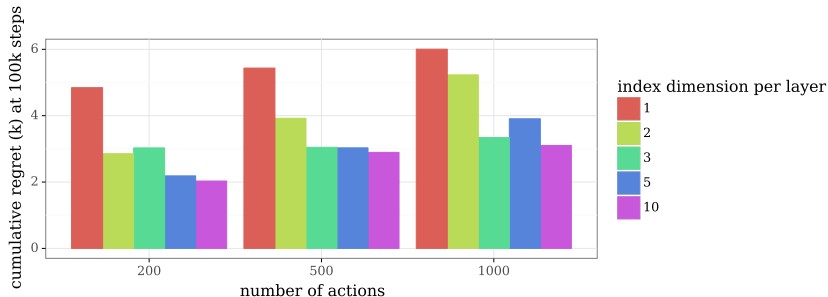

Figure 8: Performance of linear hypermodels for different index dimensions

## D    DIAGONAL LINEAR HYPERMODELS AND BAYES BY BACKPROP

An alternative approach for approximating posterior distributions for neural networks is variational methods such as Bayes by Backprop (Blundell et al., 2015). Bayes by Backprop assumes a diagonal Gaussian distribution as the variational posterior of neural network weights, which in effect uses a diagonal linear hypermodel. Its training algorithm follows the variational inference framework and aims to minimize a KL loss.

One can also train a diagonal linear hypermodel using perturbed SGD as stated in Section 2.1. Fixing the diagonal hypermodel architecture, one can ask whether perturbed SGD or whether Bayes by Backprop is a better training algorithm when used for Thompson sampling. We test these algorithms on the neural network bandit problem in Section 4.2.

We find that when training a diagonal hypermodel using perturbed SGD, base models that use an additive prior as in Section 2.5 are difficult to train. Instead, we consider base models that are a single neural network whose weights are given by $DBz + \theta$, where $z \sim N(0, I)$. The prior is encoded in $DBz$, where matrix $B$ has rows sampled from the unit hypersphere during initialization and $D$ encodes appropriate standard deviations. The learnable parameter $\theta = g_\nu(z) = \mu + Cz$

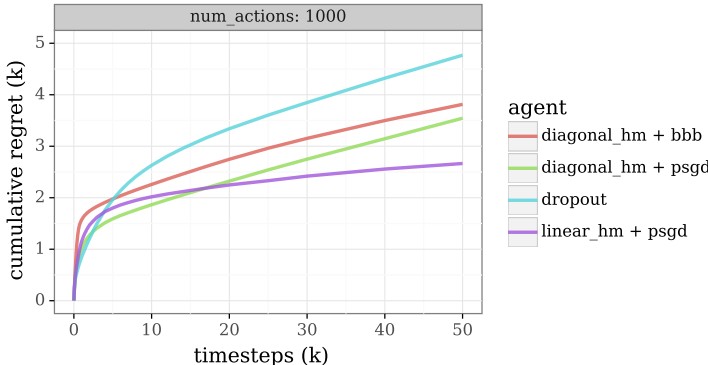

Figure 9: Compare (i) a diagonal hypermodel agent trained with perturbed SGD, (ii) a diagonal hypermodel agent trained with Bayes by Backprop, (iii) a linear hypermodel agent trained with perturbed SGD, and (iv) a dropout agent on a small neural network bandit.

where $C$ is a diagonal matrix and both $\mu$ and $C$ are initialized to zero. Initially, the weights of the base model are dominated by $DBz$, which is desirable since we want samples from the prior when there is little or no data.

Further, to make results comparable with the linear hypermodel results in Section 4.2, we increase the size of the base network for diagonal hypermodel agents so that the number of trainable parameters is approximately on the same level as that of a linear hypermodel agent in Section 4.2. Specifically, we let the base network be an MLP with 2 hidden layers of size 60.

We observed in our experiments that Bayes by Backprop does not work well with its originally proposed KL regularization. We find that we had to decrease the strength of the KL regularization by an order to get competitive performance. Further, Bayes by Backprop performs badly when the prior standard deviation of the weights is specified far from 0.1 to 0.3, which could suggest that Bayes by Backprop may only support very limited prior specifications. In Figure 9, we show the cumulative regret of the best tuned Bayes by Backprop agent.

Compared to Bayes by Backprop, perturbed SGD is easier to tune. We observed that perturbations do not make much difference in this toy example, and that regularization of base model parameters does not play a big role here as we are doing SGD. We plot the cumulative regret of the best tuned perturbed SGD agent in Figure 9. We see that given the diagonal hypermodel architecture, perturbed SGD performs slightly better than Bayes by Backprop. Both agents are worse than a linear hypermodel agent trained with perturbed SGD.

## E  DROPOUT AS A POSTERIOR APPROXIMATION FOR NEURAL NETWORKS

Another popupar approach for approximating posterior distributions for neural networks is dropout (Gal & Ghahramani, 2016). The dropout approach applies independent Bernoulli masks to the activations during training, and Gal & Ghahramani (2016) argue that applying dropout masks once again in a forward pass approximates sampling from the posterior distribution. To make the number of trainable parameters comparable to other agents, we choose the network to have 2 hidden layers of size 100. We then sweep over the probability of keeping each neuron, and find that a keeping probability of 0.5 works well. In Figure 9, we see that the performance of a tuned dropout agent is worse than the hypermodel agents.

