# OpenReview forum: "Hypermodels for Exploration"
_ICLR.cc/2020/Conference — Accept (Poster)_

### Official Review · AnonReviewer2 · 2019-10-20
**Official Blind Review #2**

**Rating:** 6

**Review:**

The authors demonstrate advantages of a linear hypermodel over an ensemble method in exploration guided by epistemic uncertainty. They perform an empirical study in the bandit setting and claim that their approach both outperforms the ensemble method and offers a significant increase in computational efficiency. The theoretical contribution is that they prove universality in the sense that an arbitrary distribution over functions can be represented by a linear hypermodel. The experiments support their claims. Some of the explanations, however, are confusing, and relations to prior work should be clarified.

Figure 3 shows a surprisingly large performance gap between the hypermodel and the ensemble method as the number of actions increases. But how about comparing linear hypermodels with different index sizes? Do we also expect asymptotic improvement as we increase the index size?

Imprecise or confusing explanations in the paper:

1) Page 2, first Q: In theory, the effectiveness of the ensemble method should converge to that of the hypermodel as the ensemble size increases. They only tried ensemble size [10, 30, 100, 300] and then concluded that linear hypermodel can be effective regardless of the size of ensembles. Why?

3) Page 3, Section 2.1, second paragraph, first sentence: Please clarify a bit more what do you mean by perturbing data? Random shuffling of the dataset in each training epoch? What does ‘response variables’ mean?

4) Page 3, Section 2.1, second paragraph, last 2 sentences about $A_t$: we guess it should be $A_t ~ N(0, I)$ if $p_z$ is unit Gaussian according to the description in this paragraph. The current text claims it is the other way around, perhaps a typo?

5) Page 3, Section 2.1, first equation: Why take the inner product between $a$ and $z$ ? How does this reflect the randomized computation (the motivation for augmented random vector $A_t$)? The objective is to maximize the log-likelihood of the prediction under the Gaussian assumption. Please clarify the assumptions about random variables $Y_t$ at the beginning of this paragraph.

6) Same place as in 5): Why regularize hypermodel parameters such that they are not too far from the initial vector? Is $\nu_0$ actually the additive prior model described in Section 2.5?

7) Page 3, Section 2.1, second equation: why multiply $|D|$ in the first term within the parentheses? Why not just $1/|D_tilde|$ to average the prediction error over the mini-batch?

8) Page 3, Section 2.1, second equation: is the cardinality of the index set $|Z_tilde|$ independent of mini-batch size? I.e. for each training data point there could be multiple models realized by multiple indices $z$

9) Page 4, Section 2.5: Why use this decomposition for training the hypermodel? If the intuition is to keep the initial weight small, what if we just simply initialize small values for $f_\theta(x)$ without decomposition?

10) Page 5, last second sentence: The notation of partition (the set notation after ‘Here,….’) is supposed to be $\hat{\mathcal{Z}}_{x^*} = \{ z\in \hat{\mathcal{Z}} | x^* in \argmax_{x} f_{g_{\nu}(z)}(x) \}$

Minor typos:

- Page 2, third paragraph: ‘…we compare their [efficacy] when used...’ ->  [efficiency] ?
- Page 2, the last paragraph before Section 2, first sentence: ‘Approaches to approximating TS and [informatino]-directed sampling...’ -> [information]

Relations to prior work:

1. Page 2: Hypernetworks (where one neural net learns to generate the weights of another net) are much older than this recent reference of 2016. One should relate this work to the original references since 1991 [FAST0-3a][FAST5][FASTMETA1-3][CO2] in section 8 of the overview http://people.idsia.ch/~juergen/deep-learning-miraculous-year-1990-1991.html

2. Intro 2nd par: dropout was first published much earlier in 1990 as the stochastic delta rule:
Hanson, S. J.(1990). A Stochastic Version of the Delta Rule, PHYSICA D,42, 265-272. See also arXiv:1808.03578, 2018.

We might improve our rating provided the comments above were addressed in a satisfactory way in the rebuttal.

Edit after rebuttal: The authors replied: "Thanks for pointing out typos and citations that we will add." But apparently in the revised PDF this did not happen.

**Experience Assessment:**

I have published in this field for several years.

**Review Assessment: Checking Correctness Of Derivations And Theory:**

I assessed the sensibility of the derivations and theory.

**Review Assessment: Checking Correctness Of Experiments:**

I assessed the sensibility of the experiments.

**Review Assessment: Thoroughness In Paper Reading:**

I read the paper at least twice and used my best judgement in assessing the paper.

---

> ### Author Response · Authors · 2019-11-07
> **response to comments of reviewer #2**
>
> Thank you very much for your review.
>
> 0) On the performance gap between linear and ensemble hypermodels.  We are working now on an experiment to see how index dimension effects performance and will add results.
>
> 1) On larger ensembles.  We observed in the neural network bandit experiments of Section 4.2 that, surprisingly, increasing the ensemble size beyond 100 does not seem to improve performance by much, if at all.  Further, computational requirements become prohibitive as we try to increase the ensemble size beyond what we have reported.  The fact that we can attain such performance with reasonable compute is the major source of advantage for linear hypermodels.
>
> 4) The statement in the paper is correct as is.  This setup makes the perturbations Gaussian.
>
> 5) Clarifying the assumptions about random variables.  As mentioned in (4) we’ve set things up so that a^\top z is Gaussian.  Hance, we are perturbing each response by Gaussian noise leading the hypermodel to approximate a posterior distribution.  Past literature on ensemble sampling and bootstrapped DQN have used similar Gaussian perturbations.  We have added a clarifying comment.
>
> 6) On regularization.  Yes, $\nu_0$ parameterizes the additive prior.  The idea is to regularize toward the prior network.  This allows the initial model to represent prior uncertainty and guide early exploration.  In the context of Thompson sampling, this induces initial randomization.
>
> 7) Why multiply by $|D|$?  This is done to calibrate the weight of the error term against that of the regularization term so that the hypermodel parameters will adapt to approximate a posterior distribution.
>
> 8) Yes, the cardinality of the index set is independent of minibatch size.  For each training data point, there are multiple models realized by multiple indices/
>
> 9) We use the decomposition of Section 2.5, page 4, because the hypermodel needs to reflect uncertainty associated with a prior distributions.  If we simply initialize to small values, we won’t get this, and for example, Thompson samples won’t adequately vary.
>
> 10) You notation looks good.  The original sentence was technically correct too but perhaps that was confusing since the first set mentioned only identified a generic partition. which is then defined by the remainder of the sentence.  We have clarified this.
>
> Thanks for pointing out typos and citations that we will add.

---

### Official Review · AnonReviewer1 · 2019-10-24
**Official Blind Review #1**

**Rating:** 3

**Review:**

The paper builds on a classical idea of sampling model parameters apart from learning them. Specifically, it combines hierarchical sampling with neural networks and proposes models that can help explore the parameter space efficiently. The proposal is evaluated appropriately.

What exactly do we mean by intelligent exploration? Is this quantified via the #samples needed or variance of sampled parameters? Or is it via regret?

The paper is clearly written and the idea makes sense. However the experiments are essentially based on simulated data. It is not entirely clear as to how this would translate to real setups.

Is it possible that the linear hypermodel is performing well because the data was generated according to a linear model in section 5?

If the baseline is a classical ensembling setup, then why not use classical performance measures to evaluate the benefit of hypermodeling? like accuracy etc. Why are we specifically talking about bandits? In other words, does the proposed hyper sampling allow for better weak learners in general as well?


**Experience Assessment:**

I have published one or two papers in this area.

**Review Assessment: Checking Correctness Of Derivations And Theory:**

I assessed the sensibility of the derivations and theory.

**Review Assessment: Checking Correctness Of Experiments:**

I carefully checked the experiments.

**Review Assessment: Thoroughness In Paper Reading:**

I read the paper thoroughly.

---

> ### Author Response · Authors · 2019-11-07
> **response to comments of reviewer #1**
>
> Thank you very much for your review.
>
> 1)Regarding what we mean by intelligent exploration: our assessment is in terms of regret and computational requirements, though you could also assess in terms of the number of samples needed and computational requirements to draw the same qualitative conclusions.
>
> 2) Regarding simulated data: We intentionally focussed on simulated data in order to carry out controlled experiments that allow for definitive conclusions.
>
> 3) On Section 5 benefiting from linearity of the model.  Section 5 does not use a linear model.
>
> 4) On assessing performance based on regret in bandits.  It is not immediately clear what metric to use to compare approximate posterior distributions and some choices may not be easy to assess in a computationally efficient manner.  Different metrics may be more or less appropriate depending on how the approximate posterior will be used.  The motivation of our research is to provide tools for efficient exploration, so exploration performance seemed like a natural metric for us.

---

> ### Author Response · Authors · 2019-11-12
> **final comment**
>
> We were surprised by this low score and haven't gained further insight into the reviewer's reasoning after responding to the initial review.
>
> It seems to us that the most significant concern raised by this reviewer was that our experiments focused on simulated data.  We would like to emphasize that this was an important choice as our intention was to carefully design controlled experiments to isolate issues and decisively answer questions we posed.  Working with real data would have diffused focus, requiring us to simultaneously address a variety of issues that would arise from working with a real data set.

---

### Official Review · AnonReviewer4 · 2019-10-28
**Official Blind Review #4**

**Rating:** 8

**Review:**

This paper investigates the possibility of using hypermodels in improving the exploration of bandit problems. By using SGD for training the hypermodel parameters, this paper introduces a computationally efficient alternative to ensemble methods. The idea of the paper is novel and interesting; however, I do have several concerns, mainly from numerical experiments that I would like the authors to address those in the rebuttal.

1) My first and the most important concern is that the numerical experiments do not evaluate different aspects of the method. There are numerous ways to check the sensitivity of your method for the choice of hyperparameters that I think could be added to the appendix. In addition to testing various values for $\sigma_p$, $\sigma_w$, and $\nu_0$, I think that multiple experiments are missing:
    i) larger neural network
    ii) I was expecting to see what would happen without additive prior. It could be one of the baselines in Figure 3. Even though (Osband et al., 2018) discuss the effect of this extension, but the use of this model is not numerically justified.
    iii) How the experiments are sensitive to the noise of the output variable? What will happen if you do not add noise?

There are also other experiments possible such as testing on a real scenario that would significantly improve the presentation of the work. This is not a requirement though.

2) I didn't get what is the purpose of the last term in the loss function defined in Section 2.1. Why you are looking preferring $\nu$ to be close to $\nu_0$?

3) P4, "it is natural to consider linear hypermodels in which parameters a and B are linearly constrained." This sentence needs to be clarified. I didn't comprehend how you are dealing with large neural network issues.

4) In Section 6, I was expecting to see a simulation showing a comparison of linear hypermodel with hypernetworks.


Minor:
* On P2, "informatino-directed" -> "informatino-directed"
* In the second paragraph of Section 2.1, it is mentioned that a hypermodel involves perturbing data. My understanding is that what is meant here by perturbing data is to add some noise to X. However, in the later formulae, there is no such thing as perturbing data. You could say that since our numerical experiments didn't show any improvement using data perturbation, we didn't include it in our notations. Please remove the confusion.
* very minor, but I would suggest using a different notation for $a$ in Sections 2.1 and 2.3 to remove any possible confusion.
* I think that the summation in computing the variance of IDS should be over $\tilde{Z}_{x^*}$.



**Experience Assessment:**

I do not know much about this area.

**Review Assessment: Checking Correctness Of Derivations And Theory:**

I did not assess the derivations or theory.

**Review Assessment: Checking Correctness Of Experiments:**

I carefully checked the experiments.

**Review Assessment: Thoroughness In Paper Reading:**

I read the paper at least twice and used my best judgement in assessing the paper.

---

> ### Author Response · Authors · 2019-11-07
> **response to comments of reviewer #4**
>
> Thank you very much for your review.
>
> 1)With regards to numerical experiments, as you suggest, we will include results in the appendix on sensitivity analysis with respect to hyperparameters.
> i) Trying larger neural networks would entail a lot of computational work and would be difficult for us to get done within the rebuttal time frame. Also, it is not clear to us that this would add insight given that the current network is already of nontrivial size.
> ii) We have added results in the appendix showing that ensembles without additive priors perform terribly.
> iii) We are running an experiment without noise and will add the results when they are available.
>
> 2)The idea here is to regularize towards the prior network.  This is essential as it induces exploration algorithms to resolve the prior uncertainty.  In the case of Thompson sampling, this is what leads to randomization of initial samples.
>
> 3) The point here is that it is easy in our framework to introduce structure to reduce the number of parameters.  For example, graphical structure associated with conditional independencies can be imposed through linear constraints.
>
> 4) This section is theoretical and offers a new fundamental and possibly surprising result on the representation power of linear hypermodels.
>
> By perturbing data, we mean adding noise to response variables (adding $\sigma_2 a^\top z$ to $y$ in the loss functions of Section 2.1.  Sorry that we did not state this clearly.  We’ve edited the wording.
>
> The notation for $a$ in Sections 2.1 and 2.3 are already consistent, so we are not sure what to change.
>
> Thank you for pointing out typos.

---

> ### Comment · AnonReviewer4 · 2019-11-15
> **The authors have responded to most questions**
>
> I got the appropriate answers to most of my questions and will increase my score to 7.
>
> I would have given a higher score if there were experiments supporting your theorem.

---

### Author Response · Authors · 2019-11-07
**ABOUT THEOREM 1**

We notice that none of the reviewers have commented on Theorem 1. Our understanding is that this novel result is a significant contribution of this paper. Specifically, it states that with neural network base models, linear hypermodels can represent essentially any probability distribution over functions with finite domain. In other words, without any additional constraints (e.g. constraints on the depth/width of the base model), linear hypermodels are essentially sufficient and hypernetworks do not offer to represent a broader range of probability distributions.

---

### Author Response · Authors · 2019-11-12
**the authors' perspective on the paper and the reviews**

We believe we have clarified items the reviewers asked about and added results requested by the reviewers.  We have not subsequently heard back from reviewers.  Before the rebuttal period closes, we'd like to offer our perspective on the paper, and in particular, why we are surprised by the current low scores.  We hope our points are clear and can justify higher scores.

In our view, the paper makes a few clear and striking points:

1) Alternative hypermodels can offer dramatic gains over ensemble hypermodels -- our experiments point to cases where the speedup is 100x or greater!

2) Alternative hypermodels can enable more intelligent exploration than is done by Thompson sampling.  We carry out experiments with information-directed sampling to illustrate this, and provide computational results for an example where regret is reduced by over 25x!

3) We prove that linear hypermodels and a sufficiently complex neural network can encode essentially any distribution over functions.

We should mention that we were so surprised by result (1) that we spent a lot of time checking and verifying.  As such, in our view this is quite significant.  Result (2) is also quite striking.  We believe result (3) represents a substantial theoretical contribution.

In retrospect, we probably could have written a paper on any one of these three results individually.  That would have allowed us to emphasize its significance.  Perhaps putting all three results in one paper was too much, possibly wore down reviewers, and effectively diminished the attention or appreciation that could be afforded to any one.

---

### Decision · Program_Chairs · 2019-12-19

**Decision:**

Accept (Poster)

**Comment:**

This paper considers ensemble of deep learning models in order to quantify their epistemic uncertainty and use this for exploration in RL. The authors first show that limiting the ensemble to a small number of models, which is typically done for computational reasons, can severely limit the approximation of the posterior, which can translate into poor learning behaviours (e.g. over-exploitation). Instead, they propose a general approach based on hypermodels which can achieve the benefits of a large ensemble of models without the computational issues. They perform experiments in the bandit setting supporting their claim. They also provide a theoretical contribution, proving that an arbitrary distribution over functions can be represented by a linear hypermodel.

The decision boundary for this paper is unclear given the confidence of reviewers and their scores. However, the tackled problem is important, and the proposed approach is sound and backed up by experiments. Most of reviewers concerns seemed to be addressed by the rebuttal, with the exception of few missing references which the authors should really consider adding. I would therefore recommend acceptance.